# FACESHOT: BRING ANY CHARACTER INTO LIFE

**Junyao Gao**[1][*]**, Yanan Sun**[2][‡]**, Fei Shen**[3]**, Xin Jiang**[3]**, Zhening Xing**[2]
**Kai Chen**[2][‡]**, Cairong Zhao**[1][‡]

[1]Tongji University, [2]Shanghai AI Laboratory, [3]Nanjing University of Science and Technology.
{junyaogao,zhaocairong}@tongji.edu.cn, {feishen,xinjiang}@njust.edu.cn,
{sunyanan,xingzhening,chenkai}@pjlab.org.cn.

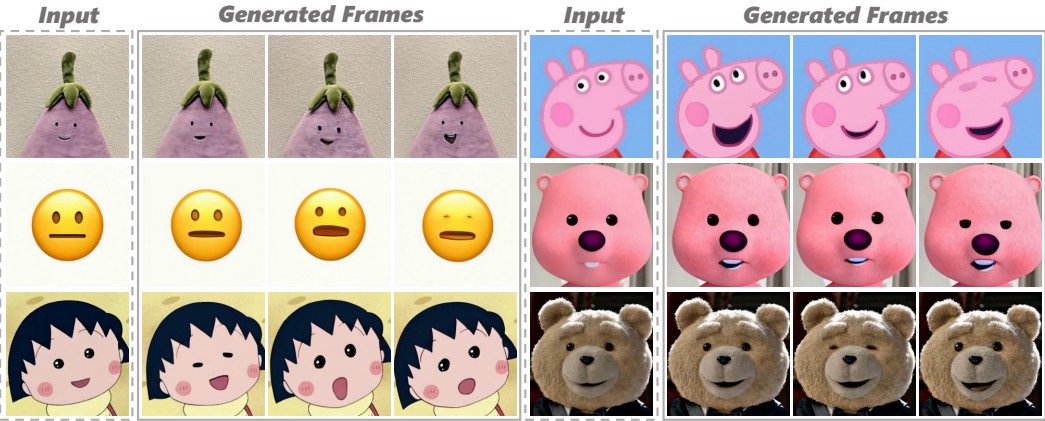

Figure 1: Visualization results of our **FaceShot**. Given any character and any driven video, FaceShot effectively captures subtle facial expressions and generates stable animations for each character. Especially for non-human characters, such as emojis and toys, FaceShot demonstrates remarkable animation capabilities.

## ABSTRACT

In this paper, we present **FaceShot**, a novel training-free portrait animation framework designed to bring any character into life from any driven video without fine-tuning or retraining. We achieve this by offering precise and robust reposed landmark sequences from an appearance-guided landmark matching module and a coordinate-based landmark retargeting module. Together, these components harness the robust semantic correspondences of latent diffusion models to produce facial motion sequence across a wide range of character types. After that, we input the landmark sequences into a pre-trained landmark-driven animation model to generate animated video. With this powerful generalization capability, FaceShot can significantly extend the application of portrait animation by breaking the limitation of realistic portrait landmark detection for any stylized character and driven video. Also, FaceShot is compatible with any landmark-driven animation model, significantly improving overall performance. Extensive experiments on our newly constructed character benchmark CharacBench confirm that FaceShot consistently surpasses state-of-the-art (SOTA) approaches across any character domain. More results are available at our project website https://faceshot2024.github.io/faceshot/.

## 1 INTRODUCTION

"I wish my toys could talk"-many people make this wish on birthdays or at Christmas, hoping for the companionship from their "imaginary friends". Achieving this usually requires a bit of "magic",

---

[*]Work done during an internship in Shanghai AI Laboratory. [‡] corresponding authors.

as seen with the talking teddy bear in *Ted*[1] or the three chipmunks in *Alvin and The Chipmunks*[2]. Behind these productions, making this "magic" a reality often requires specialized equipment and significant manual effort for character modeling and rigging. In this work, to bring any character into life for every person, we propose a novel, training-free portrait animation framework. As shown in Figure 1, even for emojis and toys, which have totally different facial appearances from humans, our proposed framework demonstrates remarkable performance in making these characters alive.

Portrait animation (Guo et al., 2024; Ma et al., 2024; Xie et al., 2024; Yang et al., 2024; Wei et al., 2024; Niu et al., 2024; Wang et al., 2021; Zeng et al., 2023) has demonstrated impressive results with the recent advancements in generative models, such as Generative Adversarial Networks (GANs) (Goodfellow et al., 2020; Donahue et al., 2016; Odena et al., 2017; Radford, 2015) and diffusion models (Rombach et al., 2022; Nichol et al., 2021; Saharia et al., 2022; Ho et al., 2020; Song et al., 2021a). However, these methods depend on facial landmark recognition, and their performance are constrained by the generalization capability of facial landmark detection models (Zhou et al., 2023; Yang et al., 2023). For non-human characters, such as emojis, animals and toys, which often exhibit significantly different facial features compared to human portraits, always resulting in landmark recognition failures due to the supervised training paradigm and limited datasets. As shown in Figure 2, the unaligned target facial features for non-human character lead to disrupted animation results; the animation models even generate a human mouth in the wrong position for a dog. In addition, Xie et al. (2024); Yang et al. (2024) indicate that these methods cannot control subtle facial motions, leading to inconsistent animation results.

To address the above limitations, we propose **FaceShot**, a novel portrait animation framework capable of animating any character from any driven video without the need for training. As demonstrated in Figure 1, FaceShot produces vivid and stable animations for various characters, particularly for non-human characters. This is achieved through three

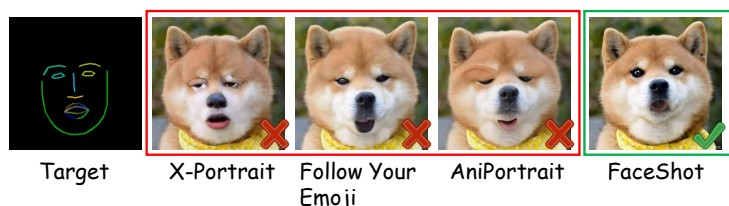

Figure 2: Visual results generated from current portrait animation methods and our FaceShot. Previous methods apparently retain the target human's appearance. In contrast, the result of FaceShot both aligns the dog's facial features and captures the target human's expression.

key components: (1) the appearance-guided landmark matching module, (2) the coordinate-based landmark retargeting module, and (3) a landmark-driven animation model.

In our landmark matching module, we inject appearance priors into diffusion features and leverage their strong semantic correspondences to match the landmarks. For the second component, we introduce a theoretical algorithm to capture subtle facial motions between frames and generate the landmark sequence aligned with the driven video. Finally, for the third component, we input the landmark sequence of the reference character into a pre-trained landmark-driven animation model to animate the character. As shown in Figure 2, FaceShot provides the reasonable animation result by offering the precise landmarks of non-human character. Furthermore, FaceShot is compatible with any landmark-driven portrait animation model as a plugin, improving their performance on non-human characters, with experimental analysis in Section 4.3.

Moreover, to address the absence of a benchmark for character animation, we establish CharacBench that contains 46 diverse characters. Qualitative and quantitative evaluations on CharacBench demonstrate that FaceShot excels in animating characters, especially in non-human characters, outperforming existing portrait animation methods. Additionally, ablation studies validate the effectiveness and superiority of our framework, providing valuable insights for the community. Furthermore, we provide the animation results of FaceShot from non-human driven videos, bringing a potential solution to the community for open-domain portrait animation.

The main contributions of this paper are as follows:

---

[1] https://en.wikipedia.org/wiki/Ted_(film)
[2] https://en.wikipedia.org/wiki/Alvin_and_the_Chipmunks_(film)

- We propose FaceShot, a novel portrait animation framework capable of animating any character from any driven video without the need for training.
- FaceShot generates precise reposed landmark sequences for any character and any driven video, bringing a potential solution to the community for open-domain portrait animation.
- FaceShot can be seamlessly integrated as a plugin with any landmark-driven animation model, further improving its performance.
- We establish CharacBench, a benchmark with diverse characters for comprehensive evaluation. Experiments on CharacBench show that FaceShot outperforms SOTA approaches.

## 2 RELATED WORK

**Portrait Animation.** Early portrait animation methods primarily relied on GANs (Goodfellow et al., 2020) to generate portrait animation through warping and rendering techniques (Drobyshev et al., 2022; Siarohin et al., 2019; Hong et al., 2022; Wang et al., 2021; Zhao & Zhang, 2022). Recent advancements in latent diffusion models (LDMs) (Rombach et al., 2022; Ramesh et al., 2022; Shen et al., 2023; Gao et al., 2024; Shen et al., 2024b; Wang et al., 2024a; Shen et al., 2024a; Li et al., 2024a; Shen & Tang, 2024) have improved the quality and efficiency of image generation. Building on this, some methods (Xie et al., 2024; Yang et al., 2024) learn the identity-free expression by constructing paired data, demonstrating impressive performance. However, current data collection pipelines face challenges in constructing paired data for diverse domains, particularly for non-human characters, which limits the generalization ability of these methods. Additionally, other approaches (Wei et al., 2024; Niu et al., 2024; Ma et al., 2024; Shen et al., 2025) tend to utilize highly scalable conditions, such as facial landmarks, to enhance motion control. Naturally, these methods depend on the facial landmark recognition, which also restricts their applicability to non-human characters. To break these limitations and bring any character into life, we focus on providing precise reposed landmark sequences within landmark-driven portrait animation in this work.

**Facial Landmark Detection** Facial landmark detection aims to detect key points in given face. Traditional methods (Cootes et al., 2000; 2001; Dollár et al., 2010; Kowalski et al., 2017) often construct a shape model for each key point and perform iterative searches to match the landmarks. With the development of deep networks, some methods (Sun et al., 2013; Zhou et al., 2013; Wu et al., 2018; 2017) select a series of coarse to fine cascaded networks to perform direct regression on the landmarks. Another trend, Huang et al. (2020); Zhou et al. (2023); Merget et al. (2018); Kumar et al. (2020) predict the heatmap of each point for indirect regression, improving the accuracy of landmark detection. Recently, Yang et al. (2023); Xu et al. (2022); Li et al. (2024b) collect larger datasets and train larger models for more generalize landmark detection. However, due to the supervised training paradigm and limited dataset, these methods are difficult to perfectly detect the landmark of non-human characters. In our paper, we turn to a training-free landmark matching module through the strong semantic correspondence and the generalization in diffusion features (Tang et al., 2023; Hedlin et al., 2024; Luo et al., 2024), aiming to provide precise landmarks for non-human characters.

**Image to Video Generation** Image to video (I2V) generation has gained significant attention in recent years due to its potential in various applications, such as image animation (Dai et al., 2023; Gong et al., 2024; Ni et al., 2023; Guo et al., 2023) and video synthesis (Blattmann et al., 2023b; Wang et al., 2024b; Ruan et al., 2023). Since diffusion models demonstrate the powerful image generation capabilities, Zhang et al. (2024); Shi et al. (2024); Xing et al. (2023); Ma et al. (2024) achieve image animation by inserting temporal layers into a pre-trained 2D UNet and fine-tuning it with video data. Furthermore, some methods (Zhang et al., 2023a; Blattmann et al., 2023a) have constructed their own I2V models and performed full training with large amounts of high-quality data, demonstrating strong competitiveness. In our work, we utilize MOFA-Video (Niu et al., 2024) as our base animation model.

## 3 METHOD

The framework of FaceShot is depicted in Figure 3. We first introduce the foundational concepts of diffusion models in Section 3.1. We then explain the three key components of our framework in detail in Section 3.2: appearance-guided landmark matching, coordinate-based landmark retargeting, and character animation model.

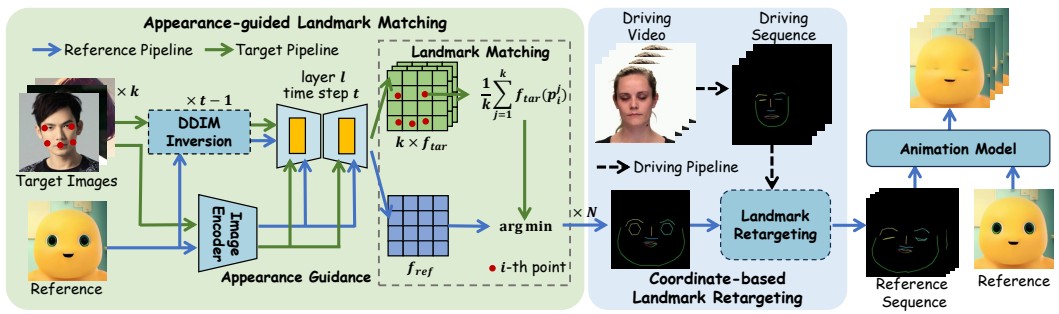

Figure 3: The FaceShot framework first generates precise facial landmarks for the reference character with appearance guidance. Next, a coordinate-based landmark retargeting module generates the landmark sequence based on driving video. Finally, this sequence is fed into an animation model to animate character.

## 3.1 PRELIMINARY

In FaceShot, we utilize Stable Diffusion (SD) (Rombach et al., 2022) as the base model for landmark matching, which consists of a Variational Auto-Encoder (VAE) (Kingma, 2013), a CLIP text encoder (Radford et al., 2021), and a denoising U-Net (Ronneberger et al., 2015). Compared to pixel-based diffusion models, SD uses the VAE encoder $\mathcal{E}$ to encode the input image $\mathbf{x}$ into a latent representation $\mathbf{z} = \mathcal{E}(\mathbf{x})$. The VAE decoder $\mathcal{D}$ then reconstructs the image by decoding the latent representation: $\mathbf{x} = \mathcal{D}(\mathbf{z})$.

To train the denoising U-Net $\epsilon_\theta$, the objective typically minimizes the Mean Square Error (MSE) loss $\mathcal{L}$ at each time step $t$, as follows:

$$\mathcal{L} = \mathbb{E}_{\mathbf{z}^t, \epsilon \sim \mathcal{N}(\mathbf{0}, \mathbf{I}), \mathbf{c}, t} \| \epsilon_\theta(\mathbf{z}^t, \mathbf{c}, t) - \epsilon^t \|^2, \tag{1}$$

where $\mathbf{z}^t = \sqrt{\bar{\alpha}^t} \mathbf{z}^0 + \sqrt{1 - \bar{\alpha}^t} \epsilon^t$ is the noisy latent at time step $t$, $\bar{\alpha}^t := \prod_{s=1}^t \alpha^s$ and $\alpha^t := 1 - \beta^t$, with $\beta^t$ represent the forward process variances. $\epsilon^t$ denotes the added Gaussian noise, and $\mathbf{c}$ is the text condition, processed by the U-Net's cross-attention module.

Moreover, the Denoising Diffusion Implicit Model (Song et al., 2021a) (DDIM) enables the inversion of the latent variable $z_0$ to $z_t$ in a deterministic manner. The formula is as follows:

$$z^t = \sqrt{\frac{\alpha^t}{\alpha^{t-1}}} z^{t-1} + + \left( \sqrt{\frac{1}{\alpha^t} - 1} - \sqrt{\frac{1}{\alpha^{t-1}} - 1} \right) \cdot \epsilon_\theta \left( z^{t-1}, t-1, c, c' \right), \tag{2}$$

where $c'$ represents the additional image prompt. In our implementation, we utilize the latent space features at $t$-th time step and $l$-th layer of U-Net from DDIM inversion for landmark matching.

## 3.2 FACESHOT: BRING ANY CHARACTER INTO LIFE

Reviewing the remarkable variance in performance caused by inaccurate or accurate facial landmarks in Figure 2. A well-generalized landmark detector is necessary for landmark-driven portrait animation model to bring any character into life. Prior landmark detectors (Yang et al., 2023; Xu et al., 2022; Zhou et al., 2023) have either curated more diverse public datasets or introduced new loss functions during training to improve the generalization of landmark detection. However, within a supervised training paradigm, these detectors struggle to generalize to non-human characters, resulting inaccurate results for portrait animation. To address this, we propose an appearance-guided landmark matching module that generalizes to any character to generate precise landmarks. In addition, to cap-

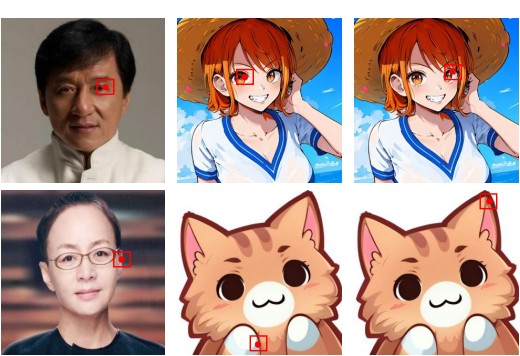

Reference image   (a) w/o appearance   (b) w/ appearance

Figure 4: Visualizations of point matching with (w/, highlighted in a red box) or without (w/o) appearance guidance using an anime diffusion model.

ture the subtle movements in driving videos, we offer a coordinate-based landmark retargeting module. Finally, a character animation model is employed as our base model to animate reference characters.

**Appearance-guided Landmark Matching.** Tang et al. (2023); Hedlin et al. (2024); Luo et al. (2024) demonstrate the strong semantic correspondence and generalization between diffusion features, where simple feature matching can map the point $p'$ on the reference image $I_{ref}$ to a semantic similar point $p$ on the target image $I_{tar}$. However, appearance discrepancies across different domains often result in mismatches, as shown in Figure 4 (a), where the points on the left eye and right ear are incorrectly matched. A natural solution is to inject prior appearance knowledge through inference using a domain-specific diffusion model. As shown in Figure 4 (b), points are correctly matched when inferred on an anime diffusion model.

Since fine-tuning a diffusion model for each reference image is costly, inspired by IP-Adapter (Ye et al., 2023), we utilize image prompts to provide appearance guidance. Specifically, we treat the reference image $I_{ref}$ and the target image $I_{tar}$ as image prompts, denoted as $c'_{ref}$ and $c'_{tar}$. We then apply the DDIM inversion process to obtain deterministic diffusion features $f_{ref}$ and $f_{tar}$ from $I_{ref}$ and $I_{tar}$ at time step $t$ and the $l$-th layer of the U-Net:

$$
\begin{aligned}
f_{tar} &= F_l(\epsilon_\theta(z_{tar}^{t-1}, t-1, c, c'_{ref})), \\
f_{ref} &= F_l(\epsilon_\theta(z_{ref}^{t-1}, t-1, c, c'_{tar})),
\end{aligned}
\tag{3}
$$

where $z_{tar}^{t-1}$ and $z_{ref}^{t-1}$ are iteratively sampled by Eq. 2 from $z_{tar}^0 = \mathcal{E}(I_{tar})$ and $z_{ref}^0 = \mathcal{E}(I_{ref})$ using the text prompt $c =$ *"a photo of a face"* and the image prompts $c'_{ref}$ and $c'_{tar}$, respectively. $F_l$ denotes the function that extracts the output feature at the $l$-th layer of the U-Net.

After obtaining the diffusion features $f_{ref} \in \mathbb{R}^{1 \times C_l \times h_l \times w_l}$ and $f_{tar} \in \mathbb{R}^{1 \times C_l \times h_l \times w_l}$, we upsample them to $f'_{ref} \in \mathbb{R}^{1 \times C_l \times H \times W}$ and $f'_{tar} \in \mathbb{R}^{1 \times C_l \times H \times W}$ to match the resolutions of $I_{ref}$ and $I_{tar}$. To improve performance and stability, we construct the average feature of the $i$-th landmark point $p_{tar}^i$ from $k$ target images to match the corresponding point $p_{ref}^i$ in the reference image, as follows:

$$
p_{ref}^i = \arg\min_{p_{ref}} d_{cos}\left(\frac{1}{k}\sum_{j=1}^{k} f_{tar}'^j(p_{tar}^{j,i}), f'_{ref}(p_{ref})\right),
\tag{4}
$$

where $f'(p) \in \mathbb{R}^{1 \times C_l}$ represents the diffusion feature vector at point $p$, $d_{cos}$ denotes the cosine distance and $p_{ref}$ refers to points in the reference feature map. Finally, we denote the matched landmark points of the reference image as $L_{ref}^0 = \{p_{ref}^i \mid i = 1, \dots, N\}$, where $N$ represents the number of facial landmarks.

**Appearance Gallery.** We introduce an appearance gallery $G = [G_e, G_m, G_n, G_{eb}, G_{fb}]$, which is a collection of five prior components—*eyes, mouth, nose, eyebrows*, and *face boundary*—across various domains, with each domain containing $k$ images. For a reference image $I_{ref}$, we reconstruct the target image as $I_{tar} = [G_e^*, G_m^*, G_n^*, G_{eb}^*, G_{fb}^*]$ by matching $I_{ref}$ with the closest domain in the appearance gallery $G$, thereby explicitly reducing the appearance discrepancy between the reference and target images, as shown in Figure 5.

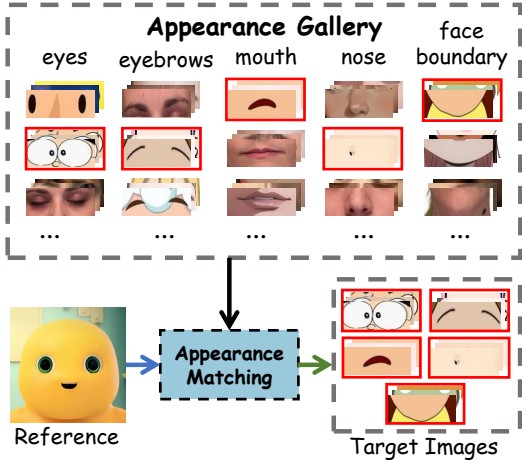

Figure 5: Illustration of our appearance gallery. We output the closest domains for each reference image to reduce the appearance discrepancy.

**Coordinate-based Landmark Retargeting.** Currently, Niu et al. (2024); Wei et al. (2024); Ma et al. (2024) utilize 3D Morphable Models (3DMM) (Booth et al., 2016) to generate the landmark sequence of the reference image by applying 3D face parameters. However, 3DMM-based methods often struggle to generalize to non-human character faces due to the limited number of high-quality 3D data and their inability to capture subtle expression movements (Retsinas et al., 2024). As shown in Figure 6, the head shapes of the 3D face are not well aligned with the input images, and subtle movements, such as eye closures, are absent in the $i$-th frame. Therefore,

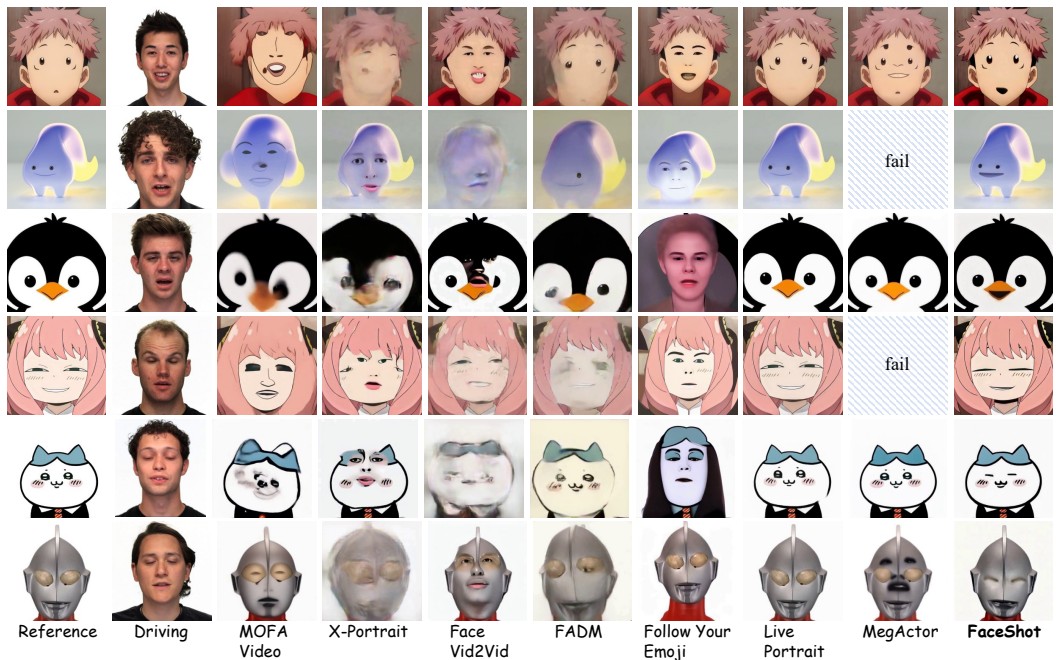

Figure 7: Qualitative comparison with SOTA portrait animation methods. Slash boxes represent that the method has fail to generate animation for this character.

we propose a coordinate-based landmark retargeting module designed to generate a retargeted landmark sequence $L_{ref}$ that can stably capture the subtle movements from driving video based on transformations in rectangular coordinate systems.

Our module consists of two stages, which respectively retarget the global motion of entire face and the local motion of different facial parts from driving sequence to reference image. In the first stage, the global motion from the 0-th to the $m$-th driving frame is defined as the translation $\Delta O_{dri} = O_{dri}^m - O_{dri}^0$ and rotation $\Delta\theta_{dri} = \theta_{dri}^m - \theta_{dri}^0$ of the corresponding global rectangular coordinate systems $(O_{dri}^0, \theta_{dri}^0)$ and $(O_{dri}^m, \theta_{dri}^m)$. Specifically, the global rectangular coordinate system is constructed by the origin $O$ and angle $\theta$,

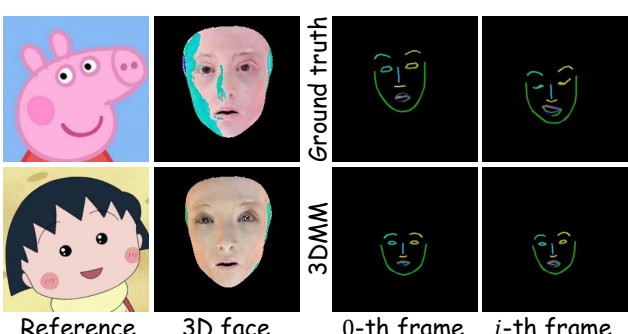

Figure 6: Visualizations of 3D face and retargeting results using 3DMM.

which are calculated from the endpoints of face boundary. Afterward, the global coordinate system of reference image at $m$-th frame can be calculated from those of 0-th frame's as follows:

$$O_{ref}^m = O_{ref}^0 + \Delta O_{dri}, \quad \theta_{ref}^m = \theta_{ref}^0 + \Delta\theta_{dri}. \tag{5}$$

Finally, we transfer the coordinates of the landmark points from $(O_{ref}^0, \theta_{ref}^0)$ to $(O_{ref}^m, \theta_{ref}^m)$, representing the global motion of the entire reference face.

In stage two, the local motion involves two processes: the relative motion and point motion, applied to five facial parts: eyes, mouth, nose, eyebrows, and face boundary. The relative motion is similar to the global motion, but the part-specific coordinate systems are calculated from the endpoints of each part. Furthermore, to constrain each part within a reasonable facial range, we scale $\Delta O_{dri}$ as $\frac{b_{ref}}{b_{dri}}\Delta O_{dri}$, where $b$ represents the distance from the origin to the boundary of each part. Next, we

model the point motion as follows:

$$p_{ref}^{m,i} = \left( \frac{p_{dri}^{m,i}[0]}{p_{dri}^{0,i}[0]} \cdot p_{ref}^{0,i}[0], \frac{p_{dri}^{m,i}[1]}{p_{dri}^{0,i}[1]} \cdot p_{ref}^{0,i}[1] \right), \qquad (6)$$

where $p_{ref}^{m,i}$ and $p_{dri}^{m,i}$ represent the coordinates in the part-specific coordinate system for the $m$-th frame and $i$-th point. This simple yet effective design enables us to capture both global and local, obvious and subtle motions into landmark sequence $L_{ref} = \{L_{ref}^j \mid j = 1, \ldots, M\}$ for any character, where $M$ represents the number of video frames.

**Character Animation Model.** After obtaining the reference landmark sequence $L_{ref}$, it can be applied to any landmark-driven animation model to animate the character portrait. Specifically, $L_{ref}$ is treated as an additional condition for the U-Net, either injected via a ControlNet-like structure (Niu et al., 2024) or incorporated directly into the latent space (Hu, 2024; Wei et al., 2024). This enables the model to precisely track the motion encoded in the landmark sequence while preserving the character's visual identity. Moreover, this flexible condition can be seamlessly extended to various architectures, enhancing scalability across diverse animation tasks.

## 4 EXPERIMENTS

### 4.1 IMPLEMENT DETAIL

In this work, we employ MOFA-Video (Niu et al., 2024), a Stable Video Diffusion (Blattmann et al., 2023a) (SVD)-based landmark-driven animation model, as our base character animation model. For appearance-guided landmark matching, we utilize Stable Diffusion v1.5 along with the pre-trained weights of IP-Adapter (Ye et al., 2023) to extract diffusion features from the images. Specifically, we set the time step $t = 301$, the U-Net layer $l = 6$, and the number of target images $k = 10$. Additionally, following MOFA-Video, we use $N = 68$ keypoints (Sagonas et al., 2016) as facial landmarks and $M = 64$ frames for animation. More details are shown in Appendix.

**Evaluation Metrics.** Following Xie et al. (2024); Ma et al. (2024), we employ four metrics to evaluate identity similarity, high- and low-level image quality and expression accuracy. Specifically, we utilize ArcFace score (Deng et al., 2019a) that calculates average cosine similarity between source and generated videos as identity similarity. We also employ HyperIQA (Zhang et al., 2023b) and LAION Aesthetic (Schuhmann et al., 2022) for evaluating image quality from low- and high-level. Moreover, we conduct the expression evaluation following the steps of Point-Tracking in MimicBench[3].

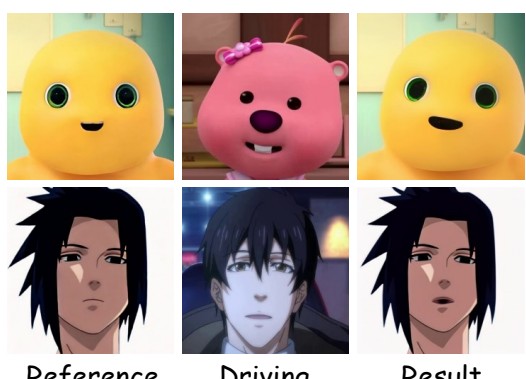

Reference     Driving     Result

Figure 8: Visualizations of character animation from non-human driving videos.

**Character Benchmark.** To comprehensively evaluate the effectiveness and generalization ability of portrait animation methods towards characters, we build CharacBench that comprises 46 characters from various domains, such as animals, emojis, toys and anime characters. Characters in CharacBench are collected from Internet by following the guideline of ensuring that the characters do not resemble human facial features as much as possible. Moreover, we consider videos of human head from RAVDESS (Livingstone & Russo, 2018) as our driving videos.

### 4.2 COMPARISON WITH SOTA METHODS

**Qualitative Results.** We compare proposed FaceShot with SOTA portrait animation methods, including MOFA-Video (Niu et al., 2024), X-Portrait (Xie et al., 2024), FaceVid2Vid (Wang et al.,

---

[3]https://github.com/open-mmlab/MimicBench

Table 1: Quantitative comparison between FaceShot and other SOTA methods on CharacBench. The best result is marked in **bold**, and the second-best performance is highlighted in underline. Symbol * indicates that there are some failure cases in these methods, we report the values of these methods *only for reference*.

| Methods | Metrics | | | | User Preference | | |
|---|---|---|---|---|---|---|---|
| | ArcFace ↑ | HyperIQA ↑ | Aesthetic ↑ | Point-Tracking ↓ | Motion ↑ | Identity ↑ | Overall ↑ |
| FaceVid2Vid | 0.525 | 33.721 | 4.267 | 6.944 | 3.58 | 3.83 | 4.52 |
| FADM | 0.633 | 39.402 | 4.522 | 6.993 | 1.93 | 2.04 | 1.96 |
| X-Portrait | 0.490 | 52.357 | 4.754 | 7.301 | 1.47 | 1.63 | 1.57 |
| Follow Your Emoji | 0.612 | 52.056 | 4.906 | 6.960 | 6.91 | 6.67 | 6.74 |
| AniPortrait* | 0.634 | 55.951 | 4.928 | 6.367 | 5.84 | 5.64 | 5.39 |
| MegaActor* | 0.613 | 40.191 | 4.855 | 7.183 | 6.53 | 6.75 | 6.26 |
| LivePortrait* | 0.893 | 53.587 | 5.092 | 7.474 | 7.33 | 7.08 | 7.11 |
| MOFA-Video | 0.695 | 52.272 | 4.952 | 14.985 | 3.27 | 3.04 | 3.18 |
| **FaceShot** | **0.848** | **53.723** | **5.036** | **6.935** | **8.14** | **8.32** | **8.27** |

2021), FADM (Zeng et al., 2023), Follow Your Emoji (Ma et al., 2024), LivePortrait (Guo et al., 2024), and MegaActor (Yang et al., 2024). Visual comparisons are presented in Figure 7, where *fail* indicates that the method was unable to generate animation for the character. As AniPortrait (Wei et al., 2024) fails with most non-human characters, we only provide its quantitative results. We observe that most methods such as MOFA-Video, X-Portrait, FaceVid2Vid and Follow Your Emoji, are influenced by the human prior in the driving video, resulting in human facial features appearing on character faces. In contrast, FaceShot effectively preserves the identity of reference characters through precise landmarks provided by our proposed appearance-guided landmark matching module. Furthermore, while most methods struggle to retarget the motions like eye closure and mouth opening, our coordinated-based landmark retargeting module enables FaceShot to capture subtle movements.

Beyond its effective character animation capability, FaceShot can also animate reference characters from non-human driving videos, extending the application of portrait animation from human-related videos to any video as shown in Fig. 8. This demonstrates its potential for open-domain portrait animations.

**Quantitative Results.** We conduct a quantitative comparison on the metrics mentioned in Section 4.1. Please note that some methods, such as Live-Portrait, MegaActor, and AniPortrait, fail to generate animations for certain characters when they are unable to detect the face. Therefore, for a fair comparison, we report the **failure rate** for these methods as follows: AniPortrait (39.13%), MegaActor (36.50%), and LivePortrait (16.67%), and we calculate their metric values on successful characters **only for reference purposes**. Based on Table 1, FaceShot demonstrates significantly superior

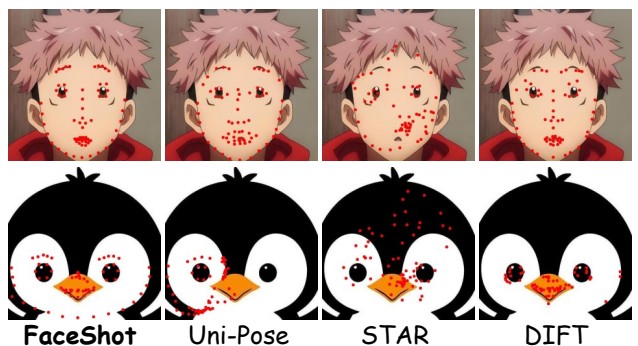

Figure 9: The visualizations of landmarks detection on different characters in CharacBench using DIFT, STAR, Uni-Pose and FaceShot.

performance across various metrics compared to other methods on CharacBench. Specifically, FaceShot achieves the highest score in terms of ArcFace (0.848), demonstrating the effectiveness of the precise landmarks generated by the appearance-guided landmark matching module in preserving facial identity. FaceShot achieves superior HyperIQA (53.723) and Aesthetic (5.036) scores, indicating better image quality. Additionally, the coordinate-based landmark retargeting module contributes to the competitive point tracking score (6.935), highlighting its ability to handle motion effectively. It is important to note that our method has achieved significant improvements across all metrics compared to the base method, MOFA-Video, further demonstrating the effectiveness of our proposed FaceShot.

**User Preference.** Additionally, we randomly selected 15 case examples and enlisted 20 volunteers to evaluate each method across three key dimensions: Motion, Identity, and Overall User Satisfaction. Volunteers ranked the animations based on these criteria, ensuring a fair and comprehensive

comparison between the methods. As shown in Table 1, FaceShot achieves the highest scores in Motion, Identity, and Overall categories, demonstrating its robust animation capabilities across diverse characters and driving videos.

## 4.3 ABLATION STUDIES

**Appearance-guided Landmark Matching.** To evaluate the effectiveness of our appearance-guided landmark matching module, we compare it with SOTA unsupervised algorithm DIFT (Tang et al., 2023) and supervised algorithms Uni-Pose (Yang et al., 2023) and STAR (Zhou et al., 2023) on CharacBench. Specifically, we recruited volunteers to annotate the landmarks of images in CharacBench as ground-truth values for calculating the corresponding Normalized Mean Error (NME) value.

Table 2: Ablation studies of our appearance-guided landmark matching module with supervised SOTA methods Uni-Pose and STAR and unsupervised method DIFT on CharacBench. Best result is marked in **bold**, and the second-best performance is highlighted in underline.

| Methods | NME ↓ | ArcFace ↑ | HyperIQA ↑ | Aesthetic ↑ |
|---|---|---|---|---|
| STAR | 24.530 | 52.849 | 0.829 | 4.989 |
| Uni-Pose | 13.731 | 53.685 | **0.851** | 5.025 |
| DIFT | 11.448 | 53.506 | 0.843 | 5.023 |
| FaceShot | **8.569** | **53.723** | 0.848 | **5.036** |

As illustrated in Table 2, FaceShot achieves the lowest NME on CharacBench. The visual landmark results are shown in Figure 9, FaceShot accurately detects the landmarks on the non-human characters, whereas others fail to match the positions of the eyes and mouth. Furthermore, we observe that Faceshot also achieves the best scores in Table 2, highlighting the necessity of precise landmarks of landmark-driven portrait animation models.

**Coordinate-based Landmark Retargeting.** To evaluate the effectiveness of our landmark retargeting module, we provide a comparison with SOTA landmark retargeting methods, including Deep3D (3DMM) (Deng et al., 2019b), Everthing's Talking (Song et al., 2021b) and FreeNet (Zhang et al., 2020), where Everthing's Talking models the Bézier Curve as the motion controller and FreeNet trains a parameterized network for retargeting.

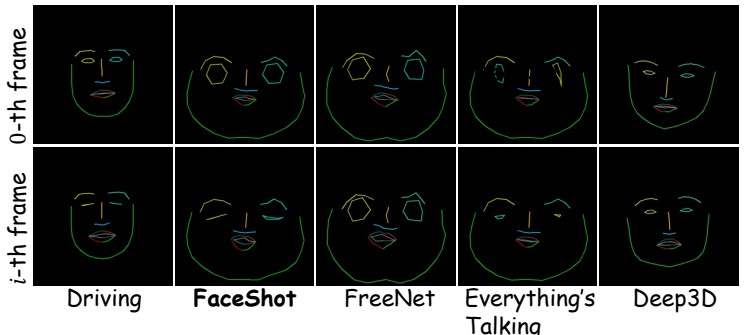

Figure 10: Visual results of landmark retargeting between FaceShot and other methods.

Table 3: Ablation studies of coordinate-based landmark retargeting module with SOTA methods Deep3D, Everthing's Talking and FreeNet. Best result is marked in **bold**.

| Metric | FaceShot | Deep3D | Everthing's Talking | FreeNet |
|---|---|---|---|---|
| Point-Tracking ↓ | **6.935** | 8.282 | 8.382 | 8.272 |

As shown in Figure 10, due to the precision loss of fitting a Bezier curve, Everthing's Talking always generates the inaccurate retargeting. FreeNet and 3DMM have strict requirements for the distribution facial features, making it unable to adapt to non-human characters. In contrast, our module can

precisely capture the subtle motion such as mouth opening, eye closure, and global face movement. Additionally, FaceShot also achieves the lowest point-tracking score in Table 3, demonstrating its effectiveness in capturing the consistent face movements.

**As a Plugin.** Experiments in Section 4.2 have demonstrated that FaceShot can significantly improve the performance of the landmark-driven method MOFA-Video (Niu et al., 2024). To further verify its effectiveness as a plugin for landmark-driven animation models, we input the reposed landmark sequences generated by FaceShot into AniPortrait's (Wei et al., 2024) pipeline. As shown in Figure 11, inaccurate landmarks in original AniPortrait often result in distortions, producing results that resemble real humans. In contrast, FaceShot can provide precise and robust landmarks for characters, leading to harmonious and stable animation results.

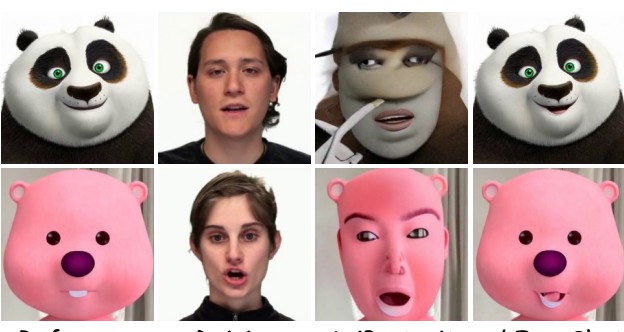

Figure 11: Visual results of AniPortrait with or without FaceShot as a plugin to generate animation results.

## 5 CONCLUSION

In this paper, we introduced FaceShot, a training-free portrait animation framework that animates any character from any driven video. By leveraging semantic correspondence in latent diffusion model features, FaceShot addresses the limitations of existing landmark-driven methods, enabling precise landmark matching and landmark retargeting. This powerful capability not only extends the application of portrait animation beyond traditional boundaries but also enhances the realism and consistency of animations in landmark-driven models. FaceShot is also compatible with any landmark-driven animation model as a plugin. Additionally, experimental results on CharacBench, a benchmark featuring diverse characters, demonstrate that FaceShot consistently outperforms current SOTA methods.

**Future Work.** Although FaceShot shows strong performance, future work could focus on enhancing appearance-guided landmark matching by refining semantic feature extraction from latent diffusion models, particularly for complex facial geometries. Furthermore, parameterizing landmark retargeting could offer more precise control over facial expressions, improving the adaptability of FaceShot across diverse character types and styles.

## ETHICS STATEMENT

In developing FaceShot, a training-free portrait animation framework that animates any characters from any driven video, we are dedicated to upholding ethical standards and promoting responsible AI use. We acknowledge potential risks, such as deepfake misuse or unauthorized media manipulation, and stress the importance of applying this technology in ways that respect privacy, consent, and individual rights. Our code will be publicly released to encourage responsible use in areas like entertainment and education, while discouraging unethical practices, including misinformation and harassment. We also advocate for continued research on safeguards and detection mechanisms to prevent misuse and ensure adherence to ethical guidelines and legal frameworks.

## ACKNOWLEDGMENTS

This project is supported by the National Key R&D Program of China (No. 2022ZD0161600) and the National Natural Science Fund of China (No. 62473286).

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

APPENDIX

# A  IMPLEMENTATION DETAILS

In FaceShot, we use a single H800 to generate animation results. And we have included a total of 46 images and 24 driving videos in CharacBench, with each video consisting of 110 to 127 frames. All videos (.mp4) and images (.jpg) are processed into a resolution of $512 \times 512$. Following MOFA-Video, for human face and driving video, we utilized Facial Alignment Network (FAN) implemented in facexlib as our annotating algorithm to detect the landmarks. And for non-human characters, we perform manual annotating as even the SOTA face landmark detection methods still fail on these characters.

# B  METHODS

**Appearance Matching in Appearance Gallery.** As mentioned in Section 3.2, the purpose of the appearance gallery is to reduce appearance discrepancies by matching the reference image to the closest target domain. In detail, the reference image is first cropped into five facial parts i.e., eyes, mouth, nose, eyebrows and face boundary as:

$$I_{ref} = [I_{ref,e}, I_{ref,m}, I_{ref,n}, I_{ref,eb}, I_{ref,fb}],$$

each facial part includes a specific number of landmarks, as listed in the Table 4:

Table 4: Specific landmark number for each facial part.

| eyes | mouth | nose | eyebrows | face boundary |
|------|-------|------|----------|---------------|
| 12 | 20 | 9 | 10 | 17 |

Next, each part is matched to the closest target domain in the appearance gallery by calculating the average CLIP image score:

$$G_p^* = \underset{j \in D}{\arg\max} \ \text{CLIP-S}(I_{ref,p}, \frac{1}{k}\sum_{i=1}^{k} I_{tar,p}^{j,i}),$$

where $p \in \{e, m, n, eb, fb\}$ and $D$ represents the domains in part $p$, $k$ denotes the number of target images in given domain and CLIP-S denotes the clip image score. Finally, the target images are formulated as $I_{tar} = [G_e^*, G_m^*, G_n^*, G_{eb}^*, G_{fb}^*]$.

**Settings in Landmark Retargeting.** The angle and the origin of the rectangular coordinate system are calculated using two endpoints $p^{e_1}$ and $p^{e_2}$ as follows:

$$O = \left( \frac{p^{e_1}[0] + p^{e_2}[0]}{2}, \frac{p^{e_1}[1] + p^{e_2}[1]}{2} \right),$$

$$\theta = \arctan\left( \frac{p^{e_2}[1] - p^{e_1}[1]}{p^{e_2}[0] - p^{e_1}[0]} \right).$$

And the indices of the endpoints within each part of every frame are fixed, as illustrated in the Figure 12. For better understanding of our coordinate-based landmark retargeting module, we provide an illustration of this module in Figure 13.

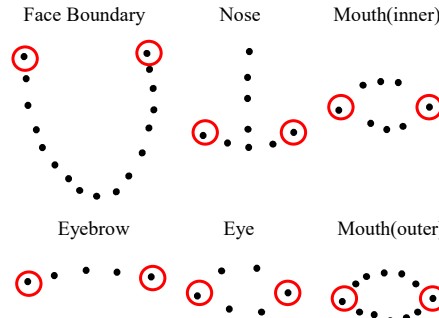

Figure 12: Illustration of the endpoints in each part, marked with a red circle.

# C  EXPERIMENTS

**Choices of Time Step $t$ and Layer $l$.** To extract the diffusion feature that best fits facial instances, we conduct detailed experiments on the selection of time steps $t$ and layer $l$ of U-Net. Specifically,

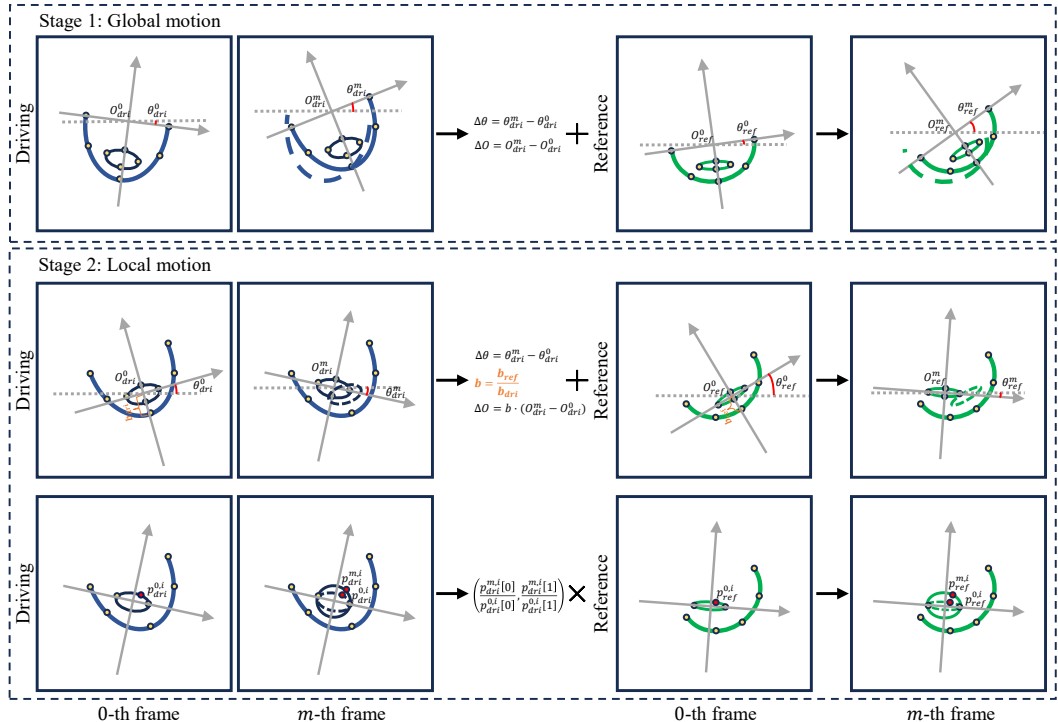

Figure 13: Illustration of our coordinate-based landmark retargeting module. Specifically, our module consists of two stages: global motion and local motion retargeting, which aim to capture global and local positional changes of the entire face and individual facial parts separately.

we test different combinations of $t$ and $l$ on 300W (Sagonas et al., 2016), a widely used facial dataset, and report NME as the quantitative result. As shown in Figure 14, we achieve the best NME value when $t = 301$ and $l = 6$, which are used as the basic settings of our paper.

**Choice of Target Number $k$.** As mentioned in Section 3.2, we use the averaged diffusion feature at the $i$-th point of $k$ target images to improve matching performance. We evaluate different values of $k = 1, 5, 10, 15, 50, 1000$ on the 300W dataset and report the NME in Table 5. Results indicate that increasing $k$ significantly improves performance when $k \leq 10$. However, for $k > 10$, the time cost increases exponentially with diminishing performance gains. Based on this observation, we set $k = 10$ for our experiments. Please note that this analysis is solely aimed at determining the number of target images. However, the features of the target images will be pre-stored as local data, which is not considered additional time overhead during inference.

**3DMM Modeling.** Following our base model MOFA-Video, we adopt Deep3D (Deng et al., 2019b) as the 3DMM method in Figure 6. Deep3D employs a deep network to predict 3D coefficients (coeff) at each frame of driven videos, instead of iterative fitting. Additionally, we

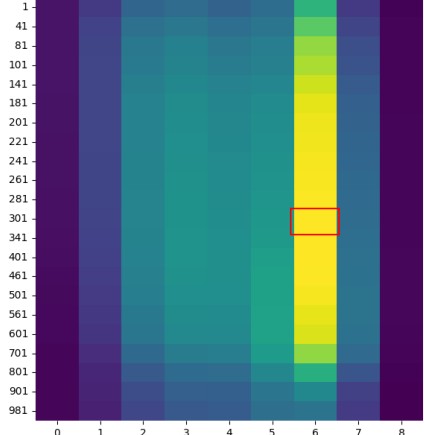

Figure 14: Heatmap of NME values for time step $t$ and layer $l$ of the U-Net.

provide the 3D modeling results of DECA (Feng et al., 2021), Deep3D and 3DDFAv2 (Guo et al., 2020) on non-human characters and driven videos in Figure 15. Our observations reveal that none of these 3DMM methods can accurately generate precise 3D models of non-human characters or capture subtle movements in driven sequences, such as eye closure.

**Visualization of Appearance Guidance.** We provide the cosine similarity distribution during inference. As depicted in Figure 16, with prior appearance knowledge, the similarity between reference points and unrelated target points becomes smaller, reducing the probability of mismatching.

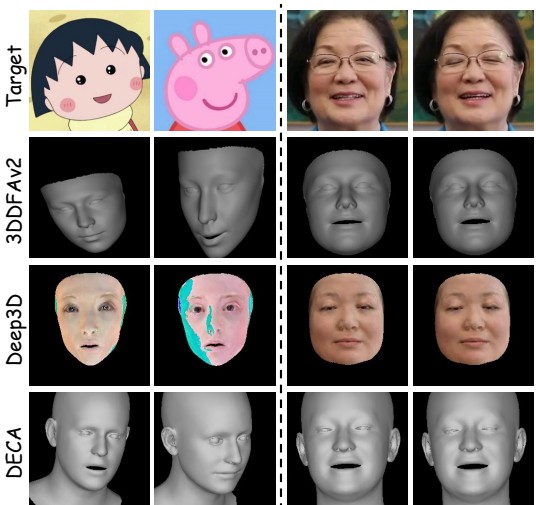

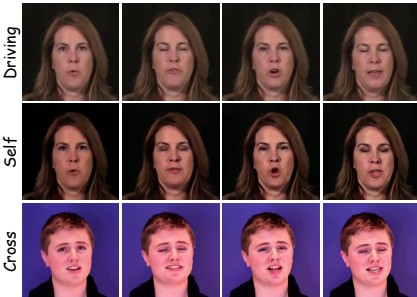

Figure 17: Visual results of landmarks on human faces.

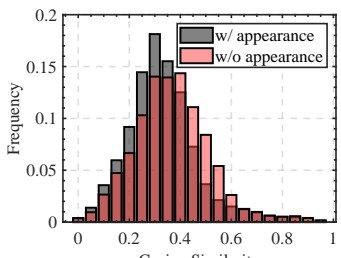

Figure 15: Modeling results of different 3DMM methods.

Figure 18: Self and cross identity driving results on HDTF.

**Human Evaluations.** To evaluate the effectiveness of FaceShot on human faces, we first provide the visual landmark results on 300W in Figure 17.

Furthermore, we present the animation results for self-identity and cross-identity driving on traditional facial video datasets HDTF (Zhang et al., 2021) in Figure 18. FaceShot perform well on these real human faces, showing its robustness.

**Time Efficiency.** We present a time analysis of each step in FaceShot for processing varying numbers of frames on a single H800 GPU, as shown in the Table 6:

Figure 16: Cosine similarity distribution with or without appearance guidance.

- *Driving Detection*: Detecting the landmark sequence of the driving video using the landmark detector from MOFA-Video. The detector used is FAN from facexlib.

- *Landmark Matching*: Detecting the target image landmarks using the appearance-guided landmark matching module. The time cost of landmark matching remains almost identical regardless of the number of frames because the matching is required only once for the target image, irrespective of the number of frames in the driving video. The 0.8-second time includes both the DDIM inversion and the argmin operation in Eq. 4.

- *Landmark Retargeting*: Retargeting landmark motion using coordinate-based landmark retargeting module. As no model parameters are required, our retargeting modules can generate precise landmark sequences with very low time cost.

Table 5: Experiments on the number $k$ of target images. Best NME result is marked in **bold**.

| Metric | 1 | 5 | 10 | 15 | 50 | 1000 |
|--------|------|------|------|------|------|------|
| NME↓ | 12.801 | 7.009 | 6.343 | 6.267 | 6.252 | **6.104** |

Table 6: Time analysis of each step in FaceShot for processing varying numbers of frames.

| frames | Driving Detection | Landmark Matching | Landmark Retargeting |
|--------|-------------------|-------------------|----------------------|
| 50 | 1.817(s) | 0.860(s) | 0.382(s) |
| 100 | 3.562(s) | 0.858(s) | 0.751(s) |

In conclusion, FaceShot introduces only a 119ms additional time overhead when used as a plugin for MOFA-Video (for 50 frames). This minimal time cost is negligible compared to the inference time of diffusion-based models (approximately 80 seconds for 50 frames). As shown in Table 7, FaceShot

achieves the low time cost among diffusion-based methods, including AniPortrait, FADM, Follow Your Emoji, MegActor, X-Portrait, and MOFA-Video.

Table 7: Time analysis SOTA methods inference on 50 frames. Symbol $^*$ represents GAN-based method.

| Methods | Time | Methods | Time |
|---|---|---|---|
| FaceVid2Vid$^*$ | 4.308(s) | MegActor | 174.189(s) |
| LivePortrait$^*$ | 4.321(s) | X-Portrait | 132.702(s) |
| AniPortrait | 99.977(s) | MOFA-Video | 79.421(s) |
| FADM | 88.368(s) | FaceShot | 79.540(s) |
| Follow Your Emoji | 112.830(s) | | |

