# OpenReview forum: "FaceShot: Bring Any Character into Life"
_ICLR.cc/2025/Conference — ICLR 2025 Poster_

### Official Review · Reviewer_x66b · 2024-10-27

**Soundness:** 3
**Presentation:** 3
**Contribution:** 3
**Rating:** 6
**Confidence:** 2

**Summary:**

The paper proposed a training free framework for animating any input character (human or non-human) from any human face driven video, by predicting facial landmarks for the input character. Extensive qualitative results showcase the effectiveness of the proposed method.

**Strengths:**

1. The proposed training-free architecture is easy to follow and and the objective as in Eq(4) is straightforward.
2. The introduced Appearance Gallery consisting of rich semantic meaningful regions benefits facial expression&motion modeling, matching and transfer.
3. The qualitative results look good and promising, especially the comparisons in Fig.12.

**Weaknesses:**

It seems that the proposed method does not significantly outperform LivePortrait in terms of both qualitative and quantitative results, as shown in Table1 and videos from Anonymous website.

**Questions:**

1. Though reported in Table 2, how long does it take for generating a single frame or the same number of frames in inference compared with other methods. e.g.  LivePortrait (with the same number of inputs)?
2. In Figure 12, why the numbers of generated landmarks with the proposed method are not consistent across different figures?
3. Some black or purple texture artifacts are witnessed on the generated images of the pink figure shown in Figure 1 (on the second row and second colume), which is more severe in its corresponding video (at the very end of the website, the 5th video from left to right). Is that because of the proposed method itself cause I haven't seen such issue on other compared methods?
4. Are there any generated videos longer than five seconds?

---

> ### Author Response · Authors · 2024-11-25
> **Rebuttal by Authors**
>
> **W1: Compared to LivePortrait**
>
> FaceShot aims to animate characters, particularly non-human characters, by providing precise landmark sequences from a training-free framework. In contrast, current portrait animation methods, such as LivePortrait, are designed based on human facial features, and therefore perform poorly on non-human characters due to the distinct facial features of non-human characters, as demonstrated in **Figure 7**. More specifically, as shown in **Figure R6**, FaceShot accurately generates results that align with the expressions in the driven frames, whereas LivePortrait fails significantly and just copies the target image as results.
>
> **Q1: Time analysis**
>
> The analysis in Table 2 is solely aimed at determining the number $k$ of reference images.
> However, the features of the reference images will be pre-stored as local data, which is not considered additional time overhead during inference.
>
> Additionally, we provide a time cost comparison for generating 50 frames with other methods, as shown in the table below:
> | FaceVid2Vid (GAN-based) | LivePortrait (GAN-based) | AniPortrait | FADM | Follow Your Emoji | MegActor | X-Portrait | MOFA-Video | FaceShot |
> | -------- | -------- | -------- | -------- | -------- | -------- | -------- | -------- | -------- |
> | 4.308(s)     | 4.321(s)     | 99.977(s)     | 88.368(s)     | 112.830(s)     | 174.189(s)     | 132.702(s)     | 79.421(s)     | 79.540(s)     |
>
> FaceShot achieves the low time cost among diffusion-based methods, including AniPortrait, FADM, Follow Your Emoji, MegActor, X-Portrait, and MOFA-Video.
>
> **Q2: Numbers of landmarks**
>
> We provide varying landmark numbers of different characters to ensure a fair evaluation of each facial part. This is because certain facial parts of some characters are missing, such as eyebrows, noses, and the facial boundary, as seen in the first case in **Figure R2**. We also present visual results for all the 68 landmarks [1] in **Figure R2** and the quantitative results in the table below:
>
> | metric | FaceShot | DIFT |Uni-Pose |STAR |
> | -------- | -------- | -------- | -------- | -------- |
> | NME $\downarrow$    | **8.569**     | 11.448     | 13.731     |  24.530     |
>
> FaceShot outperforms other methods in both qualitative and quantitative comparisons.
>
> **Q3: Black or purple texture artifacts**
>
> Texture artifacts are a common issue in diffusion-based long-term video generation, as shown in **Figure R7**, texture artifacts are also evident in other diffusion-based methods, such as X-Portrait, MOFA-Video, Follow Your Emoji and FADM. Furthermore, some methods cannot transfer motion to non-human characters, which is why they look like that they do not generate texture artifacts in animated videos, as shown in **Figure R8**. However, texture artifacts remain an unresolved issue in diffusion-based animation, and we leave this challenge for future research.
>
> **Q4: Videos longer than five seconds**
>
> Yes, we have upload generated videos longer than five seconds both in the supplementary materials and Anonymous website.
>
> [1] 300 faces in-the-wild challenge: Database and results. Image and vision computing

---

> > ### Comment · Reviewer_x66b · 2024-12-02
> >
> > Sorry for my late reply, I would like to thank authors for the detailed responses including the new attached videos.
> > I see, so FaceShot achieves the best performance among diffusion-based methods while has the fastest inference speed.
> > I would like to maintain my score at this time.
> >
> > Best, reviewer x66b.

---

> > > ### Author Response · Authors · 2024-12-02
> > >
> > > Dear Reviewer x66b,
> > >
> > > Thank you for the great efforts and valuable comments. We are encouraged to hear that your concerns have been resolved and appreciate your acknowledgment that **FaceShot achieves the best performance among diffusion-based methods while has the fastest inference speed**.  Thank you again for your efforts and please feel free to contact us if you have any further questions.
> > >
> > > Best regards,
> > >
> > > The Authors

---

> ### Author Response · Authors · 2024-11-27
> **Seeking Further Feedback**
>
> Dear Reviewer x66b:
>
> Again, we sincerely appreciate your detailed suggestions and encouragement, such as "easy to follow", "results look good and promising",  which have greatly improved our work and inspired us to research more!
>
> In our earlier response and revised manuscript, we have conducted additional experiments and provided detailed clarifications based on your questions and concerns.
>
> As we are ending the stage of the author-reviewer discussion soon, we kindly ask you to review our revised paper and our response, and to consider adjusting the scores if our response has addressed all your concerns. Otherwise, please let us know if there are any additional questions. **We would be more than happy to answer any further questions.**
>
> Best regards,
>
> The Authors

---

### Official Review · Reviewer_QeSM · 2024-11-01

**Soundness:** 3
**Presentation:** 2
**Contribution:** 3
**Rating:** 6
**Confidence:** 4

**Summary:**

This paper introduces an innovative training-free framework, FaceShot, designed to animate any human or non-human character using any driven video. The FaceShot framework is built on three key components: 1) an appearance-guided landmark matching module, 2) a relative landmark motion transfer module, and 3) a character animation model. Together, these components enable precise landmark detection and stable animation generation across diverse character types. In addition, the authors present CABench, a standardized benchmark dataset specifically created to evaluate distribution-agnostic portrait animation techniques. The techniques introduced by FaceShot represent a significant advancement in the field of open-domain portrait animation, overcoming limitations in existing methods and expanding animation capabilities to a broader range of characters

**Strengths:**

1. Broad Character Adaptability: FaceShot incorporates an "appearance-guided landmark matching module" that enables it to handle a wide variety of characters, including non-human characters like emojis, toys, and animals, without requiring retraining. This innovation significantly extends the model's applicability, allowing it to generate stable animations across diverse domains.

2. Stability in Animation Generation: FaceShot demonstrates high stability in handling non-human characters, accurately capturing subtle expression changes like eye blinks and mouth movements to produce smooth and coherent animations. This stability is mainly due to the "relative motion transfer module," which ensures continuity of facial features in the landmark sequence generation, resulting in consistent animation effects.

3. Introduction of the CABench Dataset: FaceShot not only offers a novel animation generation method but also introduces the CABench dataset, providing a standardized tool for evaluating animation performance on non-human characters. This dataset includes various character types and emotion-driven videos, offering a more representative testing environment for future research and filling a gap in the diversity of characters in current datasets.

**Weaknesses:**

Many details are missing. Without specifying the experimental setup, including the device used and the details of the training parameters (such as the learning rate), it is unclear how much time overhead is introduced by incorporating the appearance-guided landmark matching module and the relative motion transfer module. Additionally, the resolution of the generated videos has not been mentioned. See questions for details.

**Questions:**

1. How to match I_{tar} with the closest domain in the appearance gallery G, and what evaluation metric is used? This step is not indicated in the pipeline.

2. In the relative landmark motion transfer section, I have three questions. First, how to get the angle of the global rectangular coordinate system? Second, how to determine the origin and angle of the local rectangular coordinate system of each part of the face? Third, when motion occurs, the origin and angle of different local coordinate systems in the reference and target image will also change. How to deal with this?

3. CABench is considered a significant contribution of this paper, but the dataset details are missing. It would be nice to provide a detailed list of the character types and their quantities included in CABench, specifying the specific emotions and intensity distributions of the driving videos, and the number of videos for each emotion. Additionally, what is the total number of images and video frames in the dataset, and do you need any preprocessing operations performed on the dataset, including the resolution and format of the images?

4. When FaceShot is a plugin for landmark-driven animation models, will it introduce additional computation and time overhead?

5. Can Faceshot achieve real-time interaction?

6. In the pipeline, it would be nice to give some example text prompts in the  “Appearance Guided Landmark Matching” step.

---

> ### Author Response · Authors · 2024-11-25
> **Rebuttal by Authors**
>
> **W1: Experimental details**
>
> FaceShot is a training-free portrait animation framework that does not have training parameters.
> We use a single H800 to generate animation results at 512 $\times$ 512 resolution.
> Furthermore, we provide the time overhead of incorporating the appearance-guided landmark matching module and the relative motion transfer module for various frame counts, as shown in the table below:
> |    frames    | Target Matching | Motion Transfer |
> | -------- | -------- | -------- |
> | 50      | 0.860(s)     | 0.382(s)     |
> | 100      | 0.858(s)     | 0.751(s)     |
>
> *Target Matching*: Detecting the target image landmarks using the appearance-guided landmark matching module. The time cost of landmark matching remains almost identical regardless of the number of frames because the matching is required only once for the target image given any driving video. The 0.8-second time includes both the DDIM inversion and the argmin operation in Eq(4).
>
> *Motion Transfer*: Transferring landmark motion using relative landmark motion transfer module with very low time cost.
>
> **Q1: Details of appearance gallery**
>
> First, the target image is cropped into five facial parts i.e., eyes, mouth, nose, eyebrows and face boundary as:
> $$
> I_{tar} = [I_{tar, e}, I_{tar, m}, I_{tar, n}, I_{tar, eb}, I_{tar, fb}].
> $$
>
> Next, each part is matched to the closest domain in the appearance gallery by calculating the average CLIP image score for each domain:
>
> $
> G^*_p  = \underset{j \in D}{\arg \max}\ \ d( \cdot, \cdot),
> $
>
> where
> $
> d( \cdot, \cdot) = \frac{1}{k}\sum^k_{i=1}\text{CLIP-S}(I_{tar,p}, I_{ref,p}^{j,i}), \ p\in \{e, m, n, eb, fb\},
> $
> (markdown in openreview does not support some latex input, so we split this formula into two lines),
> where $D$ represents the number of domain in part $p$, $k$ denotes the number of images in given domain and CLIP-S denotes the clip image score.
> Finally, the reference image is formulated as $I_{ref} = [G_e^*, G_m^*, G_n^*, G_{eb}^*, G_{fb}^*]$.
>
> **Q2: Details of relative landmark motion transfer**
>
> For better understanding, we provide an illustration of relative landmark motion transfer in **Figure R5**. Specifically, our module consists of two stages: global motion transfer and local motion transfer.
> For global motion transfer, we focus on the overall positional changes of the entire face, represented by the discrepancy in the origin $O$ and angle $\theta$ of the rectangular coordinate systems between the $0$-th frame and the $m$-th frame.
> Next, we perform similar operations on each local facial part, but incorporating a scale factor to constrain the translation of the origin $O$.
> Finally, we use the transformation of landmark points within the corresponding rectangular coordinate system as the final local translation for each part.
>
> *How to get the angle of the global rectangular coordinate system ?*
>
> The angle of the global rectangular coordinate system at frame $m$ is calculated using two endpoints of face boundary as:
> $$
> \theta^m=\text{arctan}(\frac{p_{m,i}[1]-p_{m,0}[1]}{p_{m,i}[0]-p_{m,0}[0]}),
> $$
>
> *How to determine the origin and angle of the local rectangular coordinate system of each part of the face?*
>
> It is determined by the endpoints of each part. The origin is calculated as follows:
> $$O^m=((p_{m,0}[0]+p_{m,i}[0])/2, (p_{m,0}[1]+p_{m,i}[1])/2)$$
>
> *When motion occurs, the origin and angle of different local rectangular coordinate systems in the reference and target image will also change. How to deal with this?*
>
> As shown in **Figure R5**, the local motion is defined as the changes / discrepancies in the origin and angle of different local coordinate systems between two reference images.
> When motion occurs, we add the discrepancies of origin and angle to target image's local rectangular coordinate system as the motion transfer process.
>
> **Q3: Details of CABench**
>
> We provide the types and quantities of characters in CABench as follows:
>
> | Anime Characters (small eyes) | Anime Characters (normal eyes) | Anime Characters (big eyes) | Emojis | Animals | 3D characters | Human-like Characters | Toys |
> | -------- | -------- | -------- | -------- | -------- | -------- | -------- | -------- |
> | 8     | 8     | 5     | 4     | 7     | 6     | 5     | 3     |
>
> Additionally, we present the specific distribution of emotions and intensities in the driven videos, as shown below:
>
> | Intensity | Neutral | Calm | Happy | Sad | Angry | Fearful | Disgust | Surprised |
> | -------- | -------- | -------- | -------- | -------- | -------- | -------- | -------- | -------- |
> | normal     | 2     | 2     | 2     | 1     | 1     | 1     | 2     | 1     |
> | strong     | 1     | 1     | 1     | 2     | 2     | 2     | 1     | 2     |
>
> In CABench, we have included a total of 46 images and 24 driven videos, with each video consisting of 110 to 127 frames.
> All videos (.mp4) and images (.jpg) are processed into a resolution of 512 $\times$ 512.

---

> ### Author Response · Authors · 2024-11-25
> **Rebuttal by Authors**
>
> **Q4: Cost time analysis**
>
> FaceShot introduces only a 119ms additional time overhead when used as a plugin for MOFA-Video (for 50 frames).
> This minimal time cost is negligible compared to the inference time of diffusion-based models (approximately 80 seconds for 50 frames).
>
> Additionally, we provide the specific time costs for our landmark matching module (column 2) and motion transfer module (column 3) in the table below:
> |    frames    | Target Matching | Motion Transfer |
> | -------- | -------- | -------- |
> | 50      | 0.860(s)     | 0.382(s)     |
>
> **Q5: Real time interaction**
>
> FaceShot is a diffusion-based animation model that cannot achieve real-time interaction due to its diffusion process.
>
> **Q6: Text prompts**
>
> Following DIFT [1], we use the text prompt "a photo of a face" for each character. In this step, we provide the reference image as an additional image prompt for guiding the matching.
>
> [1] Emergent Correspondence from Image Diffusion, NIPS

---

> > ### Comment · Reviewer_QeSM · 2024-11-26
> > **further question**
> >
> > Are the endpoints randomly selected? Are these two points always used during the motion?

---

> ### Author Response · Authors · 2024-11-26
> **response for further question**
>
> The indices of the endpoints within each part of every frame are fixed, as illustrated in the **Figure R9** (in our revised pdf and supplementary material).
>
> And the endpoints are determined solely based on the landmarks of the current frame.
> However, only the endpoints of the $0$-th frame are used during the entire motion to compute the discrepancy in origin and angle for the rectangular coordinate systems between the $0$-th frame and the $m$-th frame.

---

> > ### Comment · Reviewer_QeSM · 2024-11-27
> >
> > Thanks for the clarification the authors provided and the efforts they made. I decide to maintain my rating after reading the response and other reviewers' comments.

---

> ### Author Response · Authors · 2024-11-27
> **Please let us know if your concerns have been addressed**
>
> Dear Reviewer QeSM,
>
> We hope the provided response has addressed all your concerns. If there are any additional issues or new questions, please feel free to let us know. The discussion period for author-reviewer interactions ends on December 2nd, and we would be happy to provide further clarifications or discuss any points in more detail before then.
>
> **If all concerns have been resolved to your satisfaction, we kindly request that you consider revising your initial rating.**
>
> **We sincerely appreciate your valuable feedback and look forward to your response.**
>
> Best regards,
>
> The Authors

---

### Official Review · Reviewer_fAcM · 2024-11-03

**Soundness:** 3
**Presentation:** 3
**Contribution:** 2
**Rating:** 6
**Confidence:** 5

**Summary:**

FaceShot achieves precise landmark matching and robust motion transfer by introducing an appearance-guided landmark matching module and a relative motion transfer module. The landmark matching module utilizes the semantic correspondences of a latent diffusion model to extract consistent and robust landmarks from various types of driving videos. Subsequently, the motion transfer module effectively conveys the relative motion information of these landmarks to the target character, enabling diverse and realistic character animation without needing fine-tuning or retraining.

**Strengths:**

1.FaceShot offers a potential solution for portrait animation in open domains. Traditional methods often fail with non-human characters, such as emojis, animals, and toys, due to their facial features significantly differing from humans. This results in landmark detection failures and compromised animation quality. FaceShot addresses this by providing accurate landmarks, making animations for non-human characters more reasonable and effective.
2.FaceShot can be integrated as a plugin into other landmark-based animation-driven models, enhancing its scalability across various animation tasks.
3.FaceShot achieves precise landmark matching by leveraging semantic correspondences within latent diffusion models.

**Weaknesses:**

1. I like this application scenario as it effectively utilizes the correspondence between the features and positions of diffusion for landmark detection. However, without training, the effectiveness seems limited. See Question 7.
2. The cross-domain expression driving has been effectively improved, but the pose driving remains limited. However, this is a common drawback of 2D methods. See Question 2.

**Questions:**

1. Is the purpose of the appearance gallery to match the target image with a composite reference image (from different images)?
2. What is the structure of the LandmarkMotionTransfer module? Can you describe the working principle of this part?
3. On the CABench benchmark proposed by the authors, Faceshot achieved SOTA performance. Is this a test benchmark for the same identity (character)? Besides qualitative experiments, how is the effect of cross-identity motion measured? What are the quantitative results for same-identity and cross-identity driving on traditional facial video datasets, such as voxceleb1 or 2, or HDTF?
4. Is the 3DMM using an iterative fitting algorithm based on LSFM? If so, it seems intuitive that it might not perform well on video sequences. Additionally, how many iterations does it take? Have algorithms like DECA, Deep3D, 3DDFAv2, etc., been tested for performance on video sequences?
5. Which algorithm is used for annotating the landmarks in the reference images?
6. How is the stability and robustness of landmarks prediction with training-free?
7. In Figure 7, is the poor driving effect of the eyes in the first row and fifth row due to the difficulty in detecting round eyes?
8. How is the time efficiency? What is the specific time for each step, and what is the frame rate for animation driving?
I am willing to improve my score after addressing any questions.

**Details Of Ethics Concerns:**

No ethical issues have been identified.

---

> ### Author Response · Authors · 2024-11-25
> **Rebuttal by Authors**
>
> **W1 & Q7: Effectiveness of FaceShot**
>
> The reason for the unchanged eyes is that the driving eyes do not change as shown in **Figure R3**, and not due to poor driving effect. FaceShot aims to generate precise landmarks and capture the facial motion and apply it to the other character. Although it is a training-free method, its effectiveness in detection has been demonstrated in **Figure R2**, **Figure 11** and **Table 3**. To further validate this, as shown in **Figure R4**, when the driving human closes their eyes, FaceShot accurately aligns the character's landmarks and expression, further demonstrating its effectiveness.
>
> **W2 & Q2: Pose driving limitation & Details of Landmark Motion Transfer module**
>
> For better understanding, we provide an illustration of this module in **Figure R5**. Specifically, our module consists of two stages: global motion transfer and local motion transfer.
> For global motion transfer, we focus on the overall positional changes of the entire face, represented by the discrepancy in the origin $O$ and angle $\theta$ of the rectangular coordinate systems between the $0$-th frame and the $m$-th frame.
> Next, we perform similar operations on each local facial part, but incorporating a scale factor to constrain the translation of the origin $O$.
> Finally, we use the transformation of landmark points within the corresponding rectangular coordinate system as the final local translation for each part.
>
> As you mentioned, 2D landmarks indeed have limitations in fine-grained motion driving and fail to capture emotion. We leave it as future research for better expression driving.
>
> **Q1: Purpose of the appearance gallery**
>
> The purpose of the appearance gallery is to reduce appearance discrepancies by matching the target image to the closest domain. We perform appearance matching based on five facial parts, i.e., eyes, mouth, nose, eyebrows and face boundary rather than the entire face, which results in a composite reference image format.
> Each facial part includes a specific number of landmarks, as listed in the table below:
>
> | eye | mouth | nose | eyebrows | face boundary |
> | -------- | -------- | -------- | -------- | -------- |
> | 12     | 20     | 9     | 10     | 17     |
>
> This fine-grained matching approach also minimizes discrepancies within the same domain. For example, eyes in the anime domain can have vastly different shapes.
>
> **Q3: Evaluation on HDTF**
>
> CABench is a cross-identity benchmark where the identities of the target images (characters) and driven videos (humans) are distinctly different. The effect of cross-identity is demonstrated in **Table 1** and **Figure 7**.
>
> As the voxceleb1 and voceleb2 datasets are unavailable for download anymore, we provide only the quantitative results for FaceShot and other SOTA methods in both same-identity and cross-identity scenarios on HDTF [1], as shown below.
>
> For same-identity driving:
>
> | Method              | L1 $\downarrow$     | ssim$\uparrow$  | lpips$\downarrow$  | point-tracking$\downarrow$ |
> |---------------------|--------|-------|--------|----------------|
> | Aniportrait         | 10.645 | 0.883 | 0.087  | 4.256          |
> | FaceVid2Vid         | 6.938  | 0.906 | 0.116  | 4.343          |
> | FADM                | 8.420  | 0.893 | 0.138  | 4.317          |
> | Follow your emoji   | 7.336  | 0.890 | 0.088  | 4.261          |
> | Live portrait       | 7.562  | 0.877 | 0.099  | 4.918          |
> | MegActor            | 12.338 | 0.800 | 0.173  | 4.301          |
> | X-portrait          | 7.145  | 0.887 | 0.085  | 4.358          |
> | Mofa-video          | 16.046 | 0.746 | 0.152  | 6.814          |
> | FaceShot            | 14.479 | 0.754 | 0.127  | 4.532          |
>
> For cross-identity driving:
>
> | Method              | aesthetic$\uparrow$ | iqa_score$\uparrow$ | arcface$\uparrow$ | point-tracking$\downarrow$ |
> |---------------------|-----------|-----------|---------|----------------|
> | Aniportrait         | 4.858     | 56.52     | 0.800   | 4.174          |
> | FaceVid2Vid         | 4.647     | 43.921    | 0.867   | 4.081          |
> | FADM                | 4.417     | 38.571    | 0.489   | 4.425          |
> | Follow your emoji   | 4.757     | 47.870    | 0.822   | 4.195          |
> | Live portrait       | 4.834     | 44.362    | 0.870   | 4.836          |
> | MegActor            | 4.798     | 46.094    | 0.711   | 4.493          |
> | X-portrait          | 4.766     | 49.505    | 0.820   | 3.965          |
> | Mofa-video          | 4.865     | 44.141    | 0.808   | 7.618          |
> | FaceShot            | 4.798     | 46.508    | 0.852   | 4.352          |
>
> FaceShot achieves comparable results to other state-of-the-art (SOTA) methods in both same-identity and cross-identity scenarios. **Notably, FaceShot significantly outperforms the base model, MOFA-Video, highlighting its effectiveness**.
>
> [1] Flow-Guided One-Shot Talking Face Generation With a High-Resolution Audio-Visual Dataset, CVPR

---

> ### Author Response · Authors · 2024-11-25
> **Rebuttal by Authors**
>
> **Q4: 3DMM methods**
>
> Following our base model MOFA-Video, we adopt Deep3D[2] as the 3DMM method.
> Deep3D employs a deep network to predict 3D coefficients (coeff) at each frame of driven videos, instead of iterative fitting.
>
> Additionally, we provide the 3D modeling results of DECA[3], Deep3D and 3DDFAv2[4] on non-human characters and driven videos in **Figure R1**.
> Our observations reveal that none of these 3DMM methods can accurately generate precise 3D models of non-human characters or capture subtle movements in driven sequences, such as eye closure.
>
> [2] Accurate 3D Face Reconstruction with Weakly-Supervised Learning: From Single Image to Image Set, CVPR
>
> [3] Learning an Animatable Detailed 3D Face Model from In-The-Wild Images, SIGGRAPH
>
> [4] Towards Fast, Accurate and Stable 3D Dense Face Alignment, ECCV
>
> **Q5: Algorithm for annotating the landmarks**
>
> Following MOFA-Video, for human face and driven sequences, we utilized Facial Alignment Network (FAN) implemented in facexlib as our annotating algorithm to detect the landmarks.
>
> And for non-human characters, we perform manual annotating as even the state-of-the-art face landmark detection methods (MediaPipe [5], STAR [6], UniPose [7], DIFT [8]) still fail on these characters, as shown in **Figure R2**.
>
> [5] Media-Pipe https://github.com/google-ai-edge/mediapipe
>
> [6] STAR Loss: Reducing Semantic Ambiguity in Facial Landmark Detection, CVPR
>
> [7] Uni-Pose: Detection Any Keypoints, ECCV
>
> [8] Emergent Correspondence from Image Diffusion, NIPS
>
> **Q6: Stability and robustness of our landmark matching method**
>
> As shown in **Figure 11** and **Table 3**, we demonstrate the stability and robustness of our landmark matching method on human faces.
>
> Furthermore, to validate its effectiveness on non-human characters, we manually annotate 68 landmarks [9] of characters in CABench as ground truth values and compute the NME scores to compare FaceShot with DIFT, Uni-Pose and STAR.
> As shown in the table below, FaceShot achieves the best NME score:
>
> | metric | FaceShot | DIFT |Uni-Pose |STAR |
> | -------- | -------- | -------- | -------- | -------- |
> | NME $\downarrow$    | **8.569**     | 11.448     | 13.731     |  24.530     |
>
> Moreover, the visualization on **Figure R2** also demonstrates the stability and robustness of its landmark matching performance, while other methods fail to accurately match the positions of the eyes and mouth.
>
> **Q8: Time efficiency**
>
> We present a time analysis of each step in FaceShot for processing varying numbers of frames on a single H800 GPU, as shown in the table below:
>
> |    frames    | Driven Detection | Target Matching | Motion Transfer |
> | -------- | -------- | -------- | -------- |
> | 50      | 1.817(s)     | 0.860(s)     | 0.382(s)     |
> | 100      | 3.562(s)     | 0.858(s)     | 0.751(s)     |
>
>
> *Driven Detection*: Detecting the landmark sequence of the driving video using the landmark detector from MOFA-Video. The detection method used is FAN from facexlib.
>
> *Target Matching*: Detecting the target image landmarks using the appearance-guided landmark matching module. The time cost of landmark matching remains almost identical regardless of the number of frames because the matching is required only once for the target image, irrespective of the number of frames in the driving video. The 0.8-second time includes both the DDIM inversion and the argmin operation in Eq. (4).
>
> *Motion Transfer*: Transferring landmark motion using relative landmark motion transfer module. As no model parameters are required, our transfer modules can generate precise landmark sequences with very low time cost.
>
> Following MOFA-Video, the animation driving frame rate is set to 25 frames per second.
> Lastly, as a diffusion-based method, FaceShot requires approximately 79.540 seconds to infer a 50-frame video.
>
>
> [9] 300 faces in-the-wild challenge: Database and results. Image and vision computing

---

> > ### Comment · Reviewer_fAcM · 2024-11-28
> > **Response to Authors：**
> >
> > Q5: What is the scale of manually annotated dataset? Is the Appearance Gallery annotated?
> > Q7Q8: Does the model require training to better utilize the Appearance Gallery? Was CABench involved in the training process?

---

> ### Author Response · Authors · 2024-11-27
> **Seeking Further Feedback**
>
> Dear Reviewer fAcM:
>
> Again, thank you very much for the detailed comments.
>
> In our earlier response and revised manuscript, we have conducted additional experiments and provided detailed clarifications based on your questions and concerns. As the author-reviewer discussion phase is concluding, we kindly ask you to review our revised paper and our response and consider adjusting the scores if our response has addressed all your concerns. Otherwise, please let us know if there are any other questions. **We would be more than happy to answer any further questions.**
>
> Best regards,
>
> The Authors

---

> ### Comment · Reviewer_fAcM · 2024-11-28
> **Response to Authors**
>
> W1 & Q7:
> When given a driving image with open eyes and a target image with closed eyes, does the target image remain unchanged when the driving image shows closed eyes? Or does your method require consistency between initial expressions of driving and target images for better motion transfer? In Figure R3's second row, where the human subject has relatively small eyes, does the penguin's eyes still show corresponding changes when the human's eyes open wide?
>
> Q3:
> Both qualitative and quantitative analyses show that Faceshot does not demonstrate advantages in face-specific driving experiments. Additionally, it is understandable that MOFA-video, being a general video generation model, does not have advantages in this domain.
> However, looking at the overall contribution, the paper's main value lies in cross-domain motion transfer.
>
> new question 9：
> Compared to the original GAN-based approaches, what are the advantages of using diffusion-based methods?

---

> ### Author Response · Authors · 2024-11-28
> **Response to further feedback**
>
> Dear Reviewer fAcM:
>
> Thank you for dedicating your valuable time to review our work and for carefully reading our responses.
> We will address your further questions in the following content:
>
> **W1 & Q7: When given a driving image with open eyes and a target image with closed eyes, does the target image remain unchanged when the driving image shows closed eyes? Or does your method require consistency between initial expressions of driving and target images for better motion transfer? In Figure R3's second row, where the human subject has relatively small eyes, does the penguin's eyes still show corresponding changes when the human's eyes open wide?**
>
> *When given a driving image with open eyes and a target image with closed eyes, does the target image remain unchanged when the driving image shows closed eyes？*
>
> Yes, according to Eq. (6),  the target image with closed eyes will remain unchanged when the driving image shows closed eyes.
>
> *Does your method require consistency between initial expressions of driving and target images for better motion transfer?*
>
> Yes, it is widely recognized  that consistent initial expression leads to better performance.
> As shown in **Figure R10**, most methods struggle to perform well when faced with significantly inconsistent initial expressions.
>
> *In Figure R3's second row, where the human subject has relatively small eyes, does the penguin's eyes still show corresponding changes when the human's eyes open wide?*
>
> Yes, we provide results where the driven human opens their eyes widely  in **Figure R11**.
> Also, the same motion can be easily observed for the mouth, as illustrated in lines 1–5 of **Figure 7**.
>
> **Q3:Both qualitative and quantitative analyses show that Faceshot does not demonstrate advantages in face-specific driving experiments. Additionally, it is understandable that MOFA-video, being a general video generation model, does not have advantages in this domain. However, looking at the overall contribution, the paper's main value lies in cross-domain motion transfer.**
>
> These models are trained with human datasets and perform well in face-specific driving experiments, but they fail to generalize to non-human characters.
> In contrast, FaceShot bridges the gap in the non-human domain, enabling landmark-driven animation models to bring any character to life.
>
> **Q9:Compared to the original GAN-based approaches, what are the advantages of using diffusion-based methods?**
>
> Could you kindly provide a more detailed description of Q9?
> If you are asking why FaceShot selects the diffusion-based model as its base model, it is because most recent portrait animation methods are based on diffusion models, with the community showing increased activity and engagement, resulting in a broader range of base model options.
> Also, diffusion models beat GANs on generation quality, scalability and generalization capability.
>
> **Q5 &Q7 & Q8:What is the scale of manually annotated dataset? Is the Appearance Gallery annotated? Does the model require training to better utilize the Appearance Gallery? Was CABench involved in the training process?**
>
> We manually annotated the non-human characters in both the CABench and Appearance Gallery (a total of 86 images: 46 from CABench and 40 from the Appearance Gallery) to evaluate landmark detection (as shown in the table in Q6's response) and to extract the diffusion features of reference image's landmark points, respectively.
>
> FaceShot is a training-free framework and does not require any fine-tuning.
>
>
> **Please note**: since we are unable to update the revised pdf to include the visual results at this stage, the PDFs for **Figure R10** and **Figure R11** have been uploaded to our [anonymous github repo](https://github.com/FaceShot2024/faceshot/blob/main/FaceShot_rebuttal_iclr_1.pdf).

---

> > ### Comment · Reviewer_fAcM · 2024-12-02
> > **Response to Authors：**
> >
> > Q10: Are there any trainable components or parameters in this methodological approach? If so, please provide a detailed description

---

> > > ### Author Response · Authors · 2024-12-02
> > > **Response to Reviewer fAcM**
> > >
> > > Dear Reviewer fAcM:
> > >
> > > Again, thank you for dedicating your valuable time to review our work and for carefully reading our responses. We will address your further question in the following content:
> > >
> > > **Q10:Are there any trainable components or parameters in this methodological approach? If so, please provide a detailed description?**
> > >
> > > There are **absolutely** no components or  parameters that require training in our method. FaceShot is entirely training-free that can be seamlessly integrated as a plugin with any landmark-driven animation model, further improving its performance.
> > > Moreover, FaceShot can generate precise landmark results for any character and any driven video with a low time cost, bringing a potential solution to the community for open-domain portrait animation.
> > >
> > >
> > > As the discussion phase nears its end, we would be grateful to receive your feedback and look forward to hearing from you regarding any additional comments.
> > > **We would also be grateful if you might consider raising your score for our paper based on our efforts to address your comments. We thank you again for your effort in reviewing our paper.**
> > >
> > > Best regards,
> > >
> > > The Authors

---

> ### Author Response · Authors · 2024-12-01
> **Have our responses addressed your further quetions？**
>
> Dear Reviewer fAcM:
>
> Thank you once again for your invaluable feedback on our paper.
> We have carefully addressed your further questions and conducted additional experiments to support our responses.
> As the discussion phase nears its end, we would be grateful to receive your feedback and look forward to hearing from you regarding any additional comments.
> **We would also be grateful if you might consider raising your score for our paper based on our efforts to address your comments. We thank you again for your effort in reviewing our paper.**
>
> Best regards,
>
> The Authors

---

### Author Response · Authors · 2024-11-25
**General Comments**

Thank all reviewers for reviewing and providing constructive feedbacks to our paper. We also deeply appreciate the reviewers’ acknowledgment of:
* Open-domain adaptability (reviewers fAcM, QeSM)
* Scalability as a training-free plugin (reviewers fAcM, QeSM)
* Stable and precise results for landmarks and animation (reviewers fAcM, QeSM, x66b)

Please note that, we put the rebuttal pdf both in the Appendix of revised manuscript and the supplemental materials. In the rebuttal pdf, we included 9 figures:

* Figure R1: Modeling results of different 3DMM methods.
* Figure R2: The landmark visual results on CABench.
* Figure R3: Illustration of the reasons why the eyes in the 1st and 5-th rows of Figure 7 do not change. The reason is that the driving eyes do not change instead of poor driving quality.
* Figure R4: Eye driving case, closed eyes, for the 1st and 5-th characters in Figure 7.
* Figure R5: Illustration of our relative landmark motion transfer module.
* Figure R6: Comparisons with LivePortrait.
* Figure R7: Texture artifacts in other diffusion-based methods.
* Figure R8: Illustration of why texture artifacts occur.
* Figure R9: Illustration of the endpoints in each part, marked with a red circle.

We respond to each reviewer below to address concerns. Please take a look and let us know if further clarification / discussion is needed.
Also, we will include all these discussions in the next version and release codes, models and data of FaceShot.

---

### Meta-Review · Area_Chair_cksF · 2024-12-09

**Metareview:**

All reviewers agree to accept the paper. Reviewers appreciate the novel solution, broad adaptability, and promising results. Please be sure to address the reviewers' comments in the final version.

**Additional Comments On Reviewer Discussion:**

All reviewers agree to accept the paper.

---

### Decision · Program_Chairs · 2025-01-22

Accept (Poster)